# Wastewater Irrigation: A Promising Way for Future Sustainable Agriculture and Food Security in the United Arab Emirates

Fatima Hasan Al Hamedi [1,2], Karthishwaran Kandhan [1], Yongming Liu [3], Maozhi Ren [4], Abdul Jaleel [1] and Mohammed Abdul Muhsen Alyafei [1,*]

1. Department of Integrative Agriculture, College of Agriculture and Veterinary Medicine, United Arab Emirates University, Al Ain P.O. Box 15551, United Arab Emirates; 200270204@uaeu.ac.ae (F.H.A.H.); kandhan_k@uaeu.ac.ae (K.K.); abdul.jaleel@uaeu.ac.ae (A.J.)
2. Abu Dhabi Agriculture and Food Safety Authority, Abu Dhabi P.O. Box 52150, United Arab Emirates
3. Institute of Urban Agriculture, Chinese Academy of Agricultural Sciences, Chengdu 610213, China; liuyongming01@caas.cn
4. Agricultural Science and Technology Center, Chengdu 610299, China; renmaozhi01@caas.cn
* Correspondence: mohammed.s@uaeu.ac.ae

**Abstract:** In the recent past, the production of wastewater from domestic and industrial sources steadily increased through population growth, urbanization, the Industrial Revolution, and economic development. In the world, 80% of wastewater consists of several harmful substances and hazardous chemicals that cause many deadly effects on human beings as well as ecosystems. So, the elimination of this toxic substance before discarding it into landfills is utilized as an alternative source of water which is an emerging need. Using treated wastewater for agricultural purposes is an excellent approach to rendering wastewater beneficial. As the quantity of wastewater grows, it becomes necessity to redistribute the water in a beneficial way. The rapidly increasing world population will undoubtedly increase the food demand, which directly requires more water for irrigation purposes. The rapidly increasing world population rate will undoubtedly demand an increased food production rate, which directly impacts agricultural water usage. In order to achieve sustainability in terms of agricultural water usage, alternative water resources should be explored. In this review, we tried to focus on summarizing all the leading studies in the field of wastewater utilization, the most prominent treatment methods, and a benchmarking of their technical efficiency in agriculture with special emphasis on agriculture in the marginal lands, with special emphasis on the United Arab Emirates.

**Keywords:** climate change; water scarcity; agriculture water usage; wastewater treatment

## 1. Introduction

Water is the basic need of all living organisms. In contrast, water scarcity is one of the primary life-threatening issues worldwide, and there are several policies and precautionary methodologies used to face this problem. A sufficient supply of safe and adequate water is a fundamental need, financial advancement, well-being, and securing the life of a human [1]. Hence, the measure of water promptly accessible for humans is short of one percent of the absolute water assets on the planet. Enough supplies of spotless and safe water are fundamental for financial advancement. Industrialization and the vast growth of the population lead to a far-reaching and continuous shortage of freshwater [2]. Undoubtedly, the shortage of water is a major concern. The scientific society was taking more care about eradicating this issue to make sustainable development in water management possible. There are limited methods that compensate for the shortage of freshwater, which include desalination, recycling, the development of rainfall infrastructure projects, the plantation of trees, etc. [3].

The World Water Vision Report states that the world suffers a water crisis today, which is not due to availability of too little water; instead, it is due to the insufficient water resource management that badly affects humans and the environment [4]. The exploitation of nature by humanity resulted in a great cascade of environmental problems. Humankind is challenged by many issues related to the environment, one of which is water scarcity [5]. The primary sources contributing to water scarcity are increasing population, increasing human needs, and inadequate water infrastructure or accessibility [6]. Though the world is covered with 70% water, we are unfortunate to have to tackle water scarcity (Figure 1). Only a small fraction (1%) of freshwater is accessible for life, while the remaining is trapped in glaciers [7]. This small fraction of freshwater is vital to life. Though surface water is claimed to be the principal supply of freshwater, groundwater is relied upon globally to meet human needs and agricultural purposes [8].

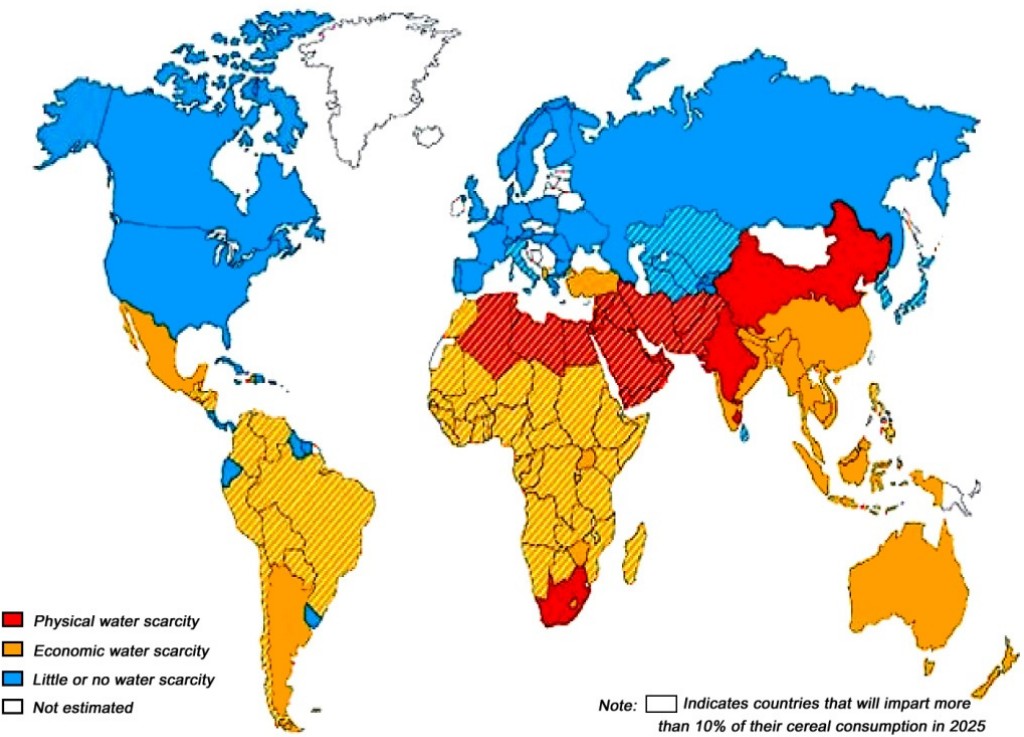

**Figure 1.** Predictable Worldwide Water Scarcity in 2025 (IWMI, http://www.cgiar.org/iwmi/home/wsmap.htm#A1).

The rapidly increasing world population rate will undoubtedly demand an increased food production rate, which directly impacts agricultural water usage [9]. This colossal water demand scenario will lead to competition for water between agriculture and other aspects of socio-economic life. With such a greater necessity of this universal solvent, according to a NASA-led study, it was reported that the world's freshwater sources are draining faster than being replenished, leading to water scarcity [10]. Despite the significant need for water, both on the surface and in groundwater, the water management protocol is unfortunately low across all areas. Quantity does not alone define water scarcity, and it also includes the quality of the water available to meet the demands of households, agriculture, industrial use, and the environment [11]. Degradation in water quality harms many global concerns, including the availability of clean drinking water and agriculture. Once contaminated, it is not easy to restore freshwater quality [12]. Freshwater pollution or contamination is due to human and animal excreta, industrial waste, chemical waste, pesticides, fertilizers, and agricultural wastes, which pose a harmful risk to the environment and living beings.

## 2. Water Scarcity

### 2.1. Water Scarcity in the Middle East and North Africa (MENA) Region

The scarcity of water is a global challenge with the rapid growth of the global population, creating an extreme burden on local water resource management. It is estimated that by 2050, the water requirement per capita in the MENA region will fall by half. With these severe consequences for the region, there is a growing scarcity of underground water and natural hydrological systems [13]. Significantly, the geographical location of the Middle East region of the world is facing tremendous water scarcity [14].

Attributable to the largest oil reserves globally within the region, most of the areas are wealthy. However, the absence of the minimum water requirements increases daily in some of the countries located in the region, including Saudi Arabia, Iraq, Iran, and Jordan, among others [15]. Among the countries, the UAE is one of the top 10 countries with water scarcity which is a major challenge for the nation. The provision of water is costly, and so, the financial resources of the nation compensate for this disbursement. Over the past three decades, the United Arab Emirates introduced a development policy to improve and sustain its economic status [16]. The increasing demand for water scarcity in the UAE is growing, and water resource utilization is restricted. Groundwater and desalinization water are essential resources for water in the UAE region. A total of 72% of water comes from groundwater, 21% comes from desalination, whereas 7% of water it is obtained from the wastewater-treated plant [17].

### 2.2. Water Scarcity in the UAE

The United Arab Emirates (UAE) uses nearly 550 L of water per person per day, making it one of the world's most water-intensive countries (250 L per person per day). According to the Gulf News, it is reported that there is a decline in the overall groundwater level, particularly in the eastern region of the emirates. This is considered one of the sustainability challenges of the UAE. It was reported that the Emirates generate more than 460,000 cubic meters of wastewater annually, increasing by 8 to 10% every year. The country plans to recycle and reuse all the wastewater to meet its needs [18]). Groundwater, desalinized water, and treated wastewater (TWW) are the primary sources of water in Abu Dhabi (TWW). In 2011, groundwater contributed 2218 million cubic meters to the Emirate's overall water use, accounting for 65% of total water use, but it is primarily a non-renewable resource. The second source of water, which accounts for 31%, is desalinated water, which has significant economic and environmental consequences [19]. TWW makes a 4% contribution. Agriculture, forestry, and landscape irrigation are the main consumers of water, accounting for 71% of all available water from all sources [20].

Water scarcity and water necessity are the biggest constraint in every nation. There is an emergency to address the increasing demand for water and food security with the growing population rate across the globe. Agriculture is a sector that consumes nearly 70% of freshwater. This higher demand for water consumption can be dealt with by recycling and reusing wastewater [21]. Appropriate and safe reuse of wastewater could prove beneficial for the environment and socio-economic development. In the UAE, about 600 million cubic meters of treated municipal wastewater is supplied, accounting for 12% of the total water supply. Around the world, about a third of wastewater treatment is used for irrigation purposes, and 20% is used to irrigate landscapes including lawns and golf courses [22].

## 3. Wastewater and Its Treatment

### 3.1. Wastewater

Typical wastewater is composed of 99.9% water while the remaining 0.1% consists of organic matter such as microorganisms (a few of which are pathogenic)—inorganic compounds—and other biodegradable organic material composed of proteins, carbohydrates, and fats as well as pathogen material [23]. Households, hospitals, and commercial buildings are significant sources of municipal wastewater, which contain significant

concentrations of macro and microelements [24]. Wastewater is usually classified into domestic sewage and non-sewage systems based on its origin. The internal sewage system is composed of wastewater generated from bathroom, public restrooms, restaurants, hotels, resorts, places of worship, sports stadiums, hospitals, and other health centers [25]. It produces enormous volumes of wastewater.

Domestic sewage water is the primary source of toxic materials and pathogenic microorganisms. The pathogens cause severe communicable and non-communicable diseases in humans, while toxic heavy metals cause critical health issues [26]. The fecal material is the primary source of the pathogen, and it potentially poses a threat to public health. The water from floods, rainwater, swimming pools, and cleaning centers is known as non-sewage water. It is adaptable for reuse and potentially less toxic [27]. Wastewater is a risk to the entire ecosystem. It imposes hazardous effects on marine life, the human race and the surrounding environment by contaminating fresh and saltwater resources. Contamination of water bodies renders the available water useless for people worldwide, creating water scarcity [28].

*3.2. Wastewater Treatment*

Wastewater is available throughout the year. Consequently, it is a trustworthy resource in regions with water scarcity or drought. Treatment of wastewater has an unusual approach to reducing water scarcity. This trend correlates to an increasing population, and it will undoubtedly contribute to an increase in the quantity of wastewater. Wastewater treatment must provide a safer environment for humans. This treatment ensures the disposal of non-toxic effluent into the water bodies without causing any pollution, or harm to aquatic life or the ecosystem [29].

The initial function of the treatment process is initiated by the natural process by which water purifies itself. Sewers collect the municipal wastewater from the residential and industrial areas and transfer the water deposits to a treatment facility that carefully treats the wastewater based on its composition (Table 1). Once the wastewater is treated, it can eradicate water scarcity, improve water resources, alleviate environmental pollution, permit sustainable nutrient recycling, and decrease fertilizer use in agriculture [30].

**Table 1.** Composition of typical domestic wastewater [31].

| Wastewater Constituents | Weak | Strong |
| --- | --- | --- |
| | All mg$^{-L-1}$ Expect Settleable Solids | |
| Alkalinity (as $CaCO_2$) | 50 | 200 |
| BOD (as $O_2$) | 100 | 300 |
| Chlorine | 30 | 100 |
| COD (as $O_2$) | 250 | 1000 |
| Suspended Solids (SS) | 100 | 350 |
| Settleable Solids (mg$^{-L-1}$) | 5 | 20 |
| Total Dissolved Solids (TDS) | 200 | 1000 |
| Total Kjeldahl Nitrogen (TKN) (as N) | 20 | 80 |
| Total Organic Carbon (TOC) (as C) | 75 | 300 |
| Total Phosphorous | 5 | 20 |

It became costly to let the wastewater go untreated and return to the environment, causing a global problem. Nearly 80% of wastewater worldwide is left untreated and surrendered to the environment, utterly wasting a potential resource. Effective treatment and efficient use of wastewater will meet human needs and prevent the contamination of groundwater, rivers, and lakes, enhancing the quality of water for future years and generations [32]. Wastewater treatment could be any of the operations physical, chemical, physicochemical, and biological. Wastewater treatment aims to eliminate toxic and hazardous chemicals and pathogens that cause diseases, and to remove or reduce other pollutants and non-desirable characteristics such as color and odor and convert it into an effluent, which could serve as an alternative to clean water [33].

When using wastewater from domestic sources, the wastewater is comprised of three different types of water: black water, grey water, and yellow water. The black water initially originates from dishwashers, toilet textures, and food waste. It includes feces, urinal waste, toilet paper, wipes, body cleaning liquids, etc. It is usually composed of highly pathogenic and toxically contaminated chemicals [34]. The wastewater originating from food fixtures and non-toilet sources is known as grey water; the wastewater from bathroom sinks, discharged water from laundry, bathtubs, and spas, and so on [35]. Yellow water is composed of urinal waste only. Comparatively, the yellow liquid is suitable for reuse among other types of wastewater [36].

### 3.2.1. Primary Treatment

The primary method of physical treatment of wastewater is the first stage of wastewater treatment in which debris and biodegradable material are removed from the wastewater. The primary method involves the sedimentation process of solid waste material, which takes place after larger waste is washed out of the water [37]. The wastewater passes through a number of filters and tanks that separate contaminants from the water to gather the sludge. After that, the collected sludge is fed into a digester, where additional processing takes place. This first batch of sludge comprises almost half of the wastewater's suspended solids. Then, the sludge undergoes secondary treatment [38].

### 3.2.2. Secondary Treatment

After the primary treatment of sewage water, secondary treatment further purifies the wastewater, and it involves three different processes, including biofiltration, aeration, and oxidation ponds. The soluble organic matter, suspended solids, and anything else that was not removed during the primary treatment are removed during the secondary treatment. Although the sewage treatment plant is made of concrete and steel, it provides a suitable environment for this biological process that occurs naturally. The consumption of the treatment plant's soluble organic matter adds to the river's dissolved oxygen balance, lake, or stream that receives the treated wastewater [39]. The first stages of secondary treatment start with the biofiltration process. Sand filters, trickling filters, and contact filters are used to remove semimetal material from the wastewater. Among the three different screens, a trickling filter is commonly used as an effective method for eliminating waste material in a small batch wastewater treatment [40].

Aeration is the second stage in secondary treatment; it is the point in the process where microorganisms are introduced to proceed with the treatment. The mixture is then aerated for up to 30 h to remove toxic material from the wastewater. In the aeration process, dissolved metals are removed through oxidation, which is the chemical amalgamation of oxygen with the majority of adverse metals, thereby removing them from the water. In this process, volatile chemicals such as ammonia, chlorine, carbon dioxide, hydrogen sulfide, iron, benzene, dichloroethylene, and perchloroethylene are removed [41].

Oxidation ponds, also known as stabilization ponds or lagoons, are the final stage of the secondary treatment. The wastewater is poured into large earthen basins, which are shallow basins used specifically for this purpose, where it is treated by a natural process using algae and, in most cases, bacteria. The development of photosynthesis allows the algae to release the oxygen required for aerobic bacteria. Sometimes, in order to reduce the required size of the pond, mechanical aerators can be used to supply additional oxygen into the system. Algae that remain in the pond effluent can be later removed by filtration or by combining chemical treatment and settling [42].

### 3.2.3. Tertiary Treatment Methods

Tertiary water treatment is the third stage of wastewater treatment, in which the suspended solids are removed after the secondary treatment of the wastewater. The biochemical oxygen demand (BOD) and total suspended solids (TSS) found in sewage are predominantly removed during primary and secondary treatment. However, this level of

treatment was proven inadequate in protecting the receiving waters or providing reusable water for domestic and industrial recycling [43]. Thus, additional steps in treatment had to be accommodated to allow for further solid and organic removal within the treatment plants to further remove nutrients and toxic materials. The methods include sedimentation, coagulation, activated carbon absorption, and filtration methods. These methods are employed in various ways to remove the poisonous content in wastewater treatment [44].

Considerable interest was shown in recent years in launching an oxidation pond drainage system because of its significance in the sewage treatment process. Organic content from drainage water, domestic water, and industrial effluents can successfully oxidized by this method [45]. Almost all mesophilic bacteria present in wastewater have oxidizing properties. The presence of indicator bacteria, bacteriophages, and viruses is undesirable and indicates the improper treatment of wastewater. Moreover, antibiotic resistance and transmissible genes are common in human pathogens in wastewater [46,47].

Oliveri et al. [48] found out that the antibiotic resistance gene of 'sul1' on the conjugative plasmid is prevalent in wastewater. When more than an acceptable microbial load is likely to grow, it is necessary to remove them. The tertiary-treated wastewater should contain less BOD and be should be free from suspended solids. The filtration process can facilitate the removal of solids. The disinfection stage in tertiary treatment determines the efficiency of disinfection in activated sludge. This is possible by mixing hypochlorite (chlorine) in the secondary treated effluent with a 5–50 ppm concentration. Thus, the tertiary treatment method will prevent many pathogens from overgrowing and decrease heavy metal concentrations in the secondary treated effluent. Quach-Cuetal. [49] demonstrated that disinfection in tertiary wastewater treatment plants significantly reduced potential resistance genes. The overall reduction was nearly 4 log10, with tertiary wastewater disinfection and filtration showing the maximum decrease.

Even after following all the primary, secondary, and tertiary techniques discussed above, success in wastewater purity, intended for agricultural reuse, is not absolute. If the suspected heavy metals can cause accumulation in the land and if the same element was traced out from the groundwater, it will prove that the particular heavy metal or pathogen is the causative substance of treated water. To avoid any contamination regarding the treated water, the soil-plant system's good management practice must be adopted [50].

The complete tertiary treatment method should involve suspending a measured quantity of antibiotic resistance genes into an available $log_{10}$ reduction (log removal value of pathogenic microorganisms (LRV)) in the solids and aqueous waste [49]. Using wastewater that was treated in soil irrigation has disadvantages because the soil and groundwater contain heavy metal accumulations of various treated water elements [45,51]. The accumulation of these heavy metals would cause an increase in the rate of soil and groundwater pollution. Additionally, soil irrigation should require a large volume of water, a lot of labor, and a large area [52].

## 4. Treated Wastewater in Agriculture

### 4.1. Global Scenario of Wastewater Use for Crop Irrigation

It is estimated that approximately 6 million ha of the area is irrigated using polluted water around the world [53]. The treated wastewater used in agriculture requires the maximum precautions to overcome consumers' health risks. The availability of microbial pathogens in sewage water might result in biological hazards for humans and vegetation. Undoubtedly, the kind of pathogen (cysts of parasites, spores of yeasts and bacteria, fungal filaments) present in any water causes a corresponding type of disease. In the UAE, this microbial agent poses a high level of potential danger for reusing wastewater. It is widely acknowledged that wastewater reuse can be an agent for water-borne diseases. Therefore, considerable attention was paid to reusing the treated wastewater in agriculture and irrigation [54].

During the Bronze Age, the earliest recycling of domestic water for agriculture was established successfully by prehistoric civilizations that used wastewater for urban settle-

ments in 3200–1100 BC. They provided an exact irrigation protocol using sewage, which was the most effective approach at the time and since went through various stages of progress [55]. Historically, according to the previous literature, the Ancient Greeks used public restrooms since the first human colonization. There was significant flushing of wastewater passing through a sewage pipe. The Romans established domestic wastewater centers in the major cities [56].

This wastewater played an essential role in the agricultural sector, influencing plant growth and promoting fertilization. Despite these applications, wastewater has many elements, that are significantly important; essential nutrients may lower the requirement for fertilizer for the vegetation and, at the same time, may increase the productivity of the crop [57]. Wastewater is a reliable source available throughout the year, unlike seasonal streams. Realizing the advantage of wastewater reuse, Germany, Scotland, and England launched wastewater-adopted agricultural farms between 1550 and 1700. After strengthening the application of wastewater for irrigation, several countries, such as the United States and Europe adopted the wastewater reuse practice during the 1800s. It has a promising possibility for reuse in agriculture as an irrigation source, especially in arid and semiarid regions. In Africa, Australia, Asia, America, Europe, and the Middle East, wastewater was reused to irrigate the soil in arid and semiarid areas [58]. Wastewater management became legal in Boston, London, and Paris. It was discovered that there was a need for large-scale wastewater treatment [59]. Paris is one of the pioneer cities of soil irrigation with wastewater in peri-urban fields. In 1872, the practice of irrigation with wastewater was expanded to four different peri-urban areas in Paris. Considerable progress in soil irrigation occurred during this same period.

The municipal wastewater disposal corporation in Australia contributed a lot to the treatment and disposal of municipal wastewater. Melbourne in 1897 established the field trial of soil irrigation with wastewater [59]. In response to the increase in mandates for food and the conservation of freshwater, wastewater management strategies were undertaken in India. The practice of using wastewater for irrigation was reported in several countries, such as Ghana, Pakistan, and Senegal [60]. Amoah et al. [61] stated that more than two hundred thousand people consume vegetables cultivated using wastewater every day in Accra, Ghana. With a share of 77 percent, Arab countries have the highest rate of reuse of treated wastewater for irrigation, compared to 14 percent in Latin America, 35 percent in Asia, and 3 percent in Africa [62]. Nothing was thought at that time about the transmission of microorganisms from wastewater to humans.

Most modern underground sewage practices are believed to have emerged in the mid-19th century due to the unhygienic disposal and health hazards that occurred in industries at that time [63]. Wastewater discharge systems were widely adopted during the 20th century with the setting up of disposal plants at major cities in the USA and UK [64]. In several areas of the world, agricultural use of wastewater increased significantly in the last 120 years to counter global water scarcity [65]. Social welfare groups reported that wastewater reuse is associated with global environmental and public health hazards. The "reuse of effluents" document was drafted by the World Health Organization (WHO) in 1973. It described health hazard control measures and wastewater treatment procedures. Methods for the irrigation of soil with the reuse of wastewater for aquaculture and agriculture were discussed in this book to sustain public health. A specific guideline is connected only with the wastewater used for irrigation and the book is not associated with the epidemiological studies.

The soil irrigation with the treated wastewater enforced a series of outbreaks that demanded the need to update the specifications in 1973. Therefore, by considering this condition, the guidelines were updated in 1989 and the replacement of conventional irrigation methods resulted by implementing risk management strategies [66]. Furthermore, the definition of acceptable microbiological standards and hazards for society because of a particular microbial disease's current situation in a country were established [67]. The specification includes several practices. Therefore, microbiological vision and safety standards of wastewater were suggested in the specifications for irrigation. Although the WHO

guidelines [66] highlighted the health protection measures to agree with the objectives, the statement on surveillance does not exist in its proforma [68]. Such surveillance documents were developed, implemented as part of the WHO wastewater specification in 2006.

Means of preventive management of wastewater constitute some of the more essential tools, such as safe use of wastewater, grey water, and excreta disposal which were described clearly in WHO's 2006 recommendations for decision makers on possible wastewater applications in agriculture. The primary purposes are to formulate regulations and standards for the management of wastewater and to consider particular aspects of every country [69,70]. Such principles are given as specifications to collect data on pathogens in irrigated fields, crops, wastewater, and eliminate potential microbial hazards. As was mentioned, one of the guidelines is to assess the control of health threats and prevent hazardous wastewater disposal.

The quality assurance requirements for using wastewater for agriculture were summarized and published in 1987. The assurance guidelines involve identifying salinity and toxicity parameters of specific ions, and wastewater use is limited. In the World Health Organization report in 1999, the FAO suggested specifications for "agriculture's care criteria and the reuse of treated waters". According to these procedures, wastewater must be adapted to the types of crops for which it is intended. Since 1992, the Environmental Protection Agency (EPA) focused primarily on the assured toxic effects of trace elements dissolved in wastewater irrigated on crops in the field. In light of the information available since the previous guidelines, the chapters on emerging chemicals and pathogens of concern, sources of information, funding alternatives, economics, research activities, public involvement, as well as acceptance and disinfection technologies were expanded in 2004 [71].

In response to global experiences, changes were made in the 2004 guidelines. To facilitate the development of recycling wastewater technologies, the United States Agency for International Development (USAID) and Environment Protection Agency (EPA) made changes in the guidelines in 2014 for Wastewater Reuse. Since the 2012 guidelines publication, many new chapters were added, such as advances in wastewater treatment technologies, the study of regional water reuse variations and best practices for engaging communities in project planning. These factors promote the expansion of water reuse activities that are both healthy and sustainable, as well as international water reuse practices. Many more experts collaborated to provide standard details, technical revisions, case studies, and updates on the book of guidelines. In several countries, the treated wastewater quality does not meet global standards because of improper treatment methods.

The impression of "desert greening" was a great motivation to extend the agricultural sector and turn the arid desert into a green paradise. Nearly 60% of water consumption is needed for the farming sector in the UAE. Despite the intense scarcity of water in agricultural resources, the UAE is mainly focusing on comfortable food security achieved by irrigation methods. In the past, all agricultural lands were irrigated using traditional irrigation methods, such as flood (furrow) and Alflaj systems. Today, modern irrigation techniques, which were introduced in the mid of 1980s, are used for localized, surface, and sprinkler irrigation. The irrigation process with reclaimed wastewater for agricultural purposes is one of the most common methods to utilize both treated and untreated wastewater. This method is comprised of restricted irrigation and non-restricted irrigation. A low quantity of water is used to grow fodder, fiber, pastures, seeds, and some nursery crops under limited irrigation. In non-restricted irrigation, a high level of treated wastewater is used to cultivate food crops that can be eaten raw. Taken together, the current research focused on the investigation of green fodder production and the usage of water, quality, and presence of heavy metal contaminants in the hydroponically produced crops using treated wastewater for irrigation and tap water irrigation.

Increased water supply and demand in the world combined with the contamination of surface and groundwater, uneven distribution, heightened pressure on the limited water resources available and recurrent droughts caused by extreme global weather patterns have placed an increased demand on advancing creative solutions to the utilization of water.

Due to these circumstances, many countries are increasingly using high-quality effluents extracted from reclamation technologies and wastewater treatment [72]. The quantity of water initially extracted from wastewater and used worldwide for irrigation of crops was shown to be much higher than conjectured, according to a detailed study. Untreated sewage threatens the health of consumers, food chain workers, and farmers, as it contains pathogens, chemical pollution, microbes, antibiotic residue, and many other contaminants. This leads to an increased concern for the environment [73].

Several approaches and technologies exist worldwide to treat, manage, and use wastewater in agriculture. The selected method tends to be specific to the farming systems and local natural resources available, as well as the products produced [74]. Reutilization of water and nutrients from localized water resources was conceived as potential food–energy–water-health (FEW–health) nexus strategies that may consume energy and increase greenhouse gas (GHG) emissions while promoting nutrient recovery and water cycling, and also providing healthy fresh foods [75].

*4.2. Benefits of Using Wastewater Reuse in Agriculture*

Treated wastewater reuse is beneficial in several chief ways in agriculture. The importance of wastewater reuse technology plays an important role in local food production and fulfills the water shortage crisis [76]. Besides alternative soil irrigation sources [77], the wastewater reuse technology has a tremendous scope for conserving available groundwater resources. There is the least chance of wastewater-sourced pollution, as it is treated before reaching the aquifer [76,77]. It lowers the pressure in freshwater sources due to the consumption of 70% of available wastewater for agriculture [78].

Uses of wastewater technology contribute positively to combating the global challenge of groundwater resource shortages. Recent research showed the remarkable effect of sewage water on crops' physiological growth. The study investigated the higher biomass in cabbage irrigated with sewage water as opposed to freshwater [79]. Safary and Hajrasoliha [80] engaged in an experiment on crops irrigated with sewage water. After seven years, they provided a clear picture that the wastewater significantly increased phosphorous, and decreased soil salinity. Rattan et al. [81] evaluated sewage effluents' beneficial effects in cereals, vegetables, and fodder crops. Singh et al. [82] also studied how sewage water significantly enhanced gram, methi, wheat, berseem, and palak. They recorded an improvement in wastewater treated soil's physicochemical properties, nutrient quality, and crop yield. The result was compared to crops irrigated with groundwater. The study conducted by Wang et al. [83] showed that sewage water increased soil macro- and micronutrient content. The increase in total carotenoid and chlorophyll content of maize treated with sewage water was recorded by Khan and Bano [84].

Wastewater irrigation is not a new concept; instead, it was in practice for centuries. Global demand for water and food led to rapid growth in the use of wastewater for irrigation. Reports suggest that knowing the need of human race, many countries show an increased rate of wastewater reuse every year [85]. Wastewater treatment plants with advanced technologies are built to ensure the quality of treated water as the composition of the water directly impacts crops. Untreated wastewater use in irrigation leads to environmental issues such as the accumulation of heavy metals, chemicals, and microbial risks in soils [86]. Adopting effective treatment methods and better management practices for the reuse of wastewater is said to bring many advantages [87]. Wastewater irrigation is beneficial since it contains high levels of nutrients, which reduces the use of chemical fertilizers [88].

Wastewater irrigation also helps conserve water resources. Comparatively, treated wastewater became a reliable source for farmers to grow crops throughout the year and again in dry areas [89]. The quality of crops grown using this treated wastewater was also found to be better. The average annual precipitation in Arab countries is less than 2% of the global average. They contain only 0.3% of the world's annual renewable freshwater resources, and so, they are challenged with a shortage of renewable freshwater resources,

leading to a state of water scarcity. Water scarcity in the Arab countries made them the largest importers of cereals from external sources to meet their food needs [90].

### 4.3. Risks of Irrigation with Untreated Wastewater

Irrigation with treated wastewater has the potential for both positive and negative environmental consequences, but with careful planning and management, it can be beneficial to the environment. When possible, treated wastewater should be reused, and disposal routes should limit any potential negative effects on the environment and human health. Excessive amounts of salt, heavy metals, minerals, toxic organic compounds (Table 2), and organic debris are the key water quality problems in relation to the chemical risks from wastewater reuse for agriculture [91]. The most appropriate treatment process for wastewater before it is utilized as irrigation water is one that produces an effluent that meets quality standards from a microbiological and chemical standpoint while requiring little operation and maintenance.

**Table 2.** Constituents present in domestic wastewater [92].

| Material | Source | Effects |
| --- | --- | --- |
| Microorganisms | Pathogenic bacteria, virus and worms eggs | Risk when bathing and eating |
| Biodegradable organic materials | Oxygen depletion in rivers, lakes, and fjords | Fish death, odors |
| Other organic materials | Detergents, pesticides, fat, oil and grease, coloring, solvents, phenols, cyanide | The toxic effect, aesthetic inconveniences, bioaccumulation in the food chain |
| Nutrients | Nitrogen, phosphorus, ammonium | Eutrophication |
| Metals | Hg, Pb, Cd, Cr, Cu, Ni | The toxic effect, bioaccumulation |
| Other inorganic materials | Acids, for example, hydrogen sulfide, bases | Corrosion, toxic effect |
| Thermal effects | Hot water | Changing living conditions for flora and fauna |
| Odor (and taste) | Hydrogen sulfide | Aesthetic inconveniences, toxic effect Radioactivity Toxic effect, accumulation |

Most wastewater treatment facilities in developing nations are either non-functional or have very low coverage, resulting in widespread water contamination and the use of low-quality water for crop irrigation, particularly in metropolitan areas. This can put people's health in danger, especially if the crops are consumed uncooked. Depending on wastewater quality and crop requirements, sedimentation, filtration, and tertiary disinfection using $ClO_2$, $O_3$, UV, or $TiO_2$ are the most suitable technologies for wastewater treatment prior to being utilized for irrigation. The importance of disinfecting wastewater during sprinkler irrigation is highlighted, as this form of irrigation increases the risk of pathogen contamination of crops, as well as bacteria and viruses that are easily spread by aerosols, in the surrounding region. Drip irrigation, as opposed to furrow irrigation, increases the possibility of direct contact between wastewater contaminated with human pathogens and the edible parts of plants, posing significant health concerns to consumers of wastewater-irrigated food.

The likelihood and magnitude of their negative effects are determined by their concentration, solubility, and inherent toxicity, as well as the rate and frequency of wastewater application, the type of crop and target yields, inherent soil properties and conditions, aquifer vulnerability, climatic conditions, and technology level, as well as the farmers' social and economic status [93]. Operators of wastewater treatment plants should work together to ensure that the effluent quality characteristics match the irrigation water requirements of each type of agricultural crop. End-users should also be educated on best practices for wastewater and effluent reuse. However, in recent years, agricultural wastewater to vegetation was the last resort because it poses an increased risk to human health and the environment [94].

### 4.4. Health and Environmental Risks of Waste Water Irrigation

Plant nutrients and organic materials can be found in wastewater. Agriculture benefits from the use of partially treated and untreated wastewater for irrigation, but it may pose health concerns to humans. However, if wastewater elements such as potentially hazardous microorganisms, antibiotic-resistant bacteria, and toxic or biologically disruptive substances are not effectively regulated, they can threaten human health. Despite these health concerns, wastewater treatment is uncommon in many nations [15]. Farmers must be informed about the hazards and best practices in this activity in order to benefit from wastewater in agriculture in a way that is safe for their health and the environment. Simultaneously, a number of risk concerns in wastewater reuse were identified; some have short-term consequences, while others have long-term consequences that worsen with ongoing wastewater use [95]. Risk knowledge, which includes the handling of wastewater-irrigated crops and vegetables, could potentially be a concern. The finest example of good practice is to use solely treated wastewater whenever possible to save both freshwater and health and environmental dangers. A greater understanding of health concerns may influence on-farm practices, reducing farmers' exposure to pathogens in wastewater and the concentration of pathogens on farm products, resulting in a lower risk of adverse health effects for both farmers and consumers. Wastewater reuse in aquaculture was also linked to negative health effects. The WHO guidelines on the reuse of wastewater in agriculture have paved the way for human health protection. It is possible to produce a higher crop yield with wastewater effluents without posing any risks if proper precautions are taken.

The guidelines advocate using a multi-barrier approach to protect farmers and farm families who come into contact with wastewater that is being reused. It is also designed to protect the food chain at crucial control or access points, particularly in arid and semi-arid regions where wastewater reuse, whether treated or untreated, is common. Another measure is continuous monitoring of effluent quality or wastewater that will be reused, in order to ensure compliance with country-specific requirements (if applicable) or WHO minimum regulations. To minimize clogging of soil pores and irrigation system emitters in the absence of complete treatment technology, wastewater must be treated by settling and/or filtering. In 2006 [95], WHO issued a set of guidelines for the safe reuse of wastewater in agriculture, which included treatment and non-treatment measures and covered the full food chain. Intestinal protozoans (Giardia and Crysptospridium) and bacterial pathogens (Giardia and Crysptospridium) are public health concerns around the world. HAV, HEV, adenovirus, and rotavirus are waterborne viruses that offer the greatest risk of transmission through wastewater reuse [94]. As a result, health and environmental authorities consider the use of microbiological markers of fecal contamination to be the most reliable means of monitoring water quality and the functioning of water treatment systems.

### 4.5. Recent Guidelines for the Safe Reuse of Wastewater Irrigation

Wastewater reuse grew in popularity around the world, prompting several countries to adopt local legislation to regulate water quality for reuse in order to reduce health and environmental hazards. Guidelines for the Safe Use of Wastewater in Agriculture must optimize public health advantages while also allowing for the efficient use of limited resources in agriculture. In the absence of relevant epidemiological data at the time of their development, previous quality recommendations and standards for wastewater reuse were based primarily on microbiological criteria. The goal was to eradicate any potential health risks provided by the presence of feces bacteria; therefore, all pathogenic fecal organisms were removed. The minimal bacterial concentrations that could be detected in the trash by routine monitoring or that could be obtained in practice by currently available treatment processes were included in the standard set. Many areas' surface water would fail to fulfill WHO fecal coliform guidelines for unrestricted irrigation.

The WHO guidelines adopted a broader focus on health-based targets, expressed in disability-adjusted life years (DALYs), which allows for the comparison of hazards and diseases, as well as the quantification of risk and the effectiveness of different pathogen

barriers in risk reduction towards the targets, via risk modeling (Figure 2). According to the kind of crop to be irrigated, guidelines for the quality of wastewater to be utilized for unrestricted agricultural irrigation, including that of salad and vegetable crops eaten raw, generally established both specific standards and minimal treatment requirements. Support for alternatives to wastewater treatment and frequent, but often unachievable, recommendations such as crop limits in the food chain was long needed [60]. In contrast to those nations grappling with unplanned reuse and in desperate need of WHO support, most countries investing in planned reuse have their own reuse criteria. Additional on- or off-farm safety measures are recommended by the WHO [94] in those geographical locations where public health cannot be protected with proper wastewater treatment. Unfortunately, such a broad and flexible approach did not translate into easy "global" rules, as the previous WHO edition, [69], particularly in light of pathogenic hazards, which the WHO prioritizes in support of low- and lower-middle-income nations.

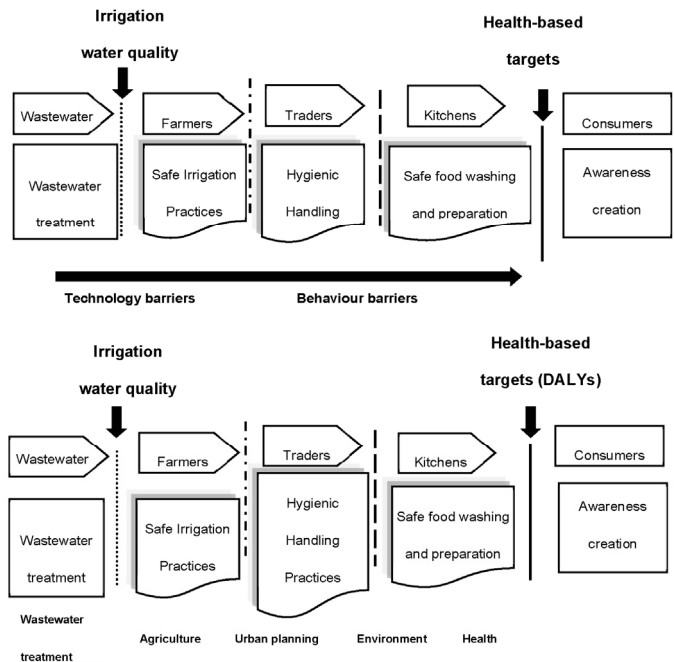

**Figure 2.** The WHO's 2006 wastewater uses guidelines.

However, agencies, officials, and others charged with managing wastewater (reuse) in Sub-Saharan Africa, Latin America, or South (East) Asia expressed concern that the four-volume 2006 guidelines are too data-intensive or complex to comprehend, while policymakers and practicing engineers in more advanced institutions that can treat wastewater for direct reuse struggle to translate the guidelines into numerical thresholds that are (for them) easy to understand [70]. To reduce pathogen concentrations in effluents to WHO suggested microbiological guideline values, tertiary treatment (e.g., filtration and/or disinfection) may be required in some circumstances. Prior to identifying acceptable risk management techniques, it is necessary to define adequate health-based targets when developing realistic recommendations for using wastewater in agriculture.

## 5. Challenges for Sustainable Agriculture in the UAE

Though the agriculture sector is not the mainstay in the Arab region, approximately thirty percent of the farmers are engaged in food and agriculture industries. Any scarcity of water resources may lead to security issues and political instability in the country. The majority of provisions, such as sugar, oils, grains, and animal products are imported from other regions to meet the needs in Arab countries. In the world, there are variations in rates of water consumption; the UAE alone accounts for the highest per capita consumption of 500 L of water per day. As Arab regions are frequently exposed to vulnerable climate

change, frequent droughts and water scarcity are common. Therefore, the UAE must educate youth, invest money in energy-efficient technology for wastewater reuse and desalination of seawater, and conserve water in aquifers.

At the national level, the food self-sufficiency ratio was stood at just 9.9% in Qatar while 29.45% in Gulf (GCC) and 86.84% in Sudan, and 80.8% in Nile valley countries. The total wastewater production was about 12.2 billion $m^3$/year in 2013, and 6.13 billion $m^3$/year volumes of untreated wastewater are discharged, which is accounted for about 51% of the total wastewater produced in the Arab region [62]. As an outcome, the UAE aims to reduce water demand, lift the water productivity index to 110 dollars per cubic meter, and reuse treated wastewater. The actual fear is that demand in consumption of desalinated water produced through the thermal process increases by an average of 4203 gallons could cause the sudden extinction of fossil fuel. Thus, the UAE focused on investing money in alternative energy-efficient ways. Despite laymen's ignorance about financial pressure from the oil crisis due to its decline in global oil price, the UAE government is approving some innovative projects.

In the UAE, approximately 60% of the groundwater is used for irrigation. As many groundwater resources are drying up, water resource advisors are determined to intensify the re-use of treated wastewater for irrigation. Just 5% of treated water is reused in Abu Dhabi. The total amount of well water established for irrigation was about 11,500 cubic meter in the Emirates. Although 20% of these water wells already dried up, as irrigated agriculture consumes 80 percent of well water across the country. Therefore, the overall commitment to sustain the groundwater resources was the national importance of the UAE [89]. By 2017, Emirate generated over 460 million cubic meters of wastewater. The report also highlighted that the Government noticed the wastewater generation increased by 8 to 10% annually. The water sector is regulated by several government agencies such as ADWEA, ADSSC, and EAD. In 2020, the plan was that over 360 million cubic meters of treated wastewater would be recycled and reused for irrigation.

The United Arab Emirates is faced with unparalleled water resource problems, including high costs of supplying drinking water, high salinity levels in developed groundwater, a lack of groundwater supplies, and restricted water re-use. With the growing demand for water, there was more significant pressure on the country's water infrastructure. The most challenging issue is establishing water efficiency in the UAE is the need of the hour. In 2019, the government allocated 1.6 billion dollars to water and energy projects. However, the UAE environment 2030 report forecasted that both brackish and groundwater sources from the aquifer system would dry up in fifty years unless precautionary steps were taken [90].

## 6. Wastewater in UAE

There is a growing increase in wastewater quantities in Arab countries; the aggressive expansion of the wastewater processing units covers most of the inhabited areas. The UAE is already practicing the use of re-treated water for daily services. Abu Dhabi practiced centralized domestic and municipal wastewater treatment procedures since 1973. The Emirate has the ambition to develop an excellent wastewater procedure. There are 32 wastewater treatment plants, dispersed equally between the Eastern and Western regions. The quality of the treated wastewater is a vital aspect to consider as treated water for agriculture purposes. This should prevent host transmission of pathogens.

Nowadays, the UAE produced an increasing quantity of wastewater and solid waste from the domestic and industrial sectors; this discharged material is commonly creating many hazardous and health issues. The wastewater generated in the UAE measures up to 500 million cubic meters a year, out of which 289 million cubic meters are treated and 248 million cubic meters of the treated volume are repurposed. Treated wastewater from Abu Dhabi and Dubai in 2017 accounted for about 45% of the total treated wastewater, and Sharjah contributed 17%. The rest of the UAE contributed only 38% (Figure 3).

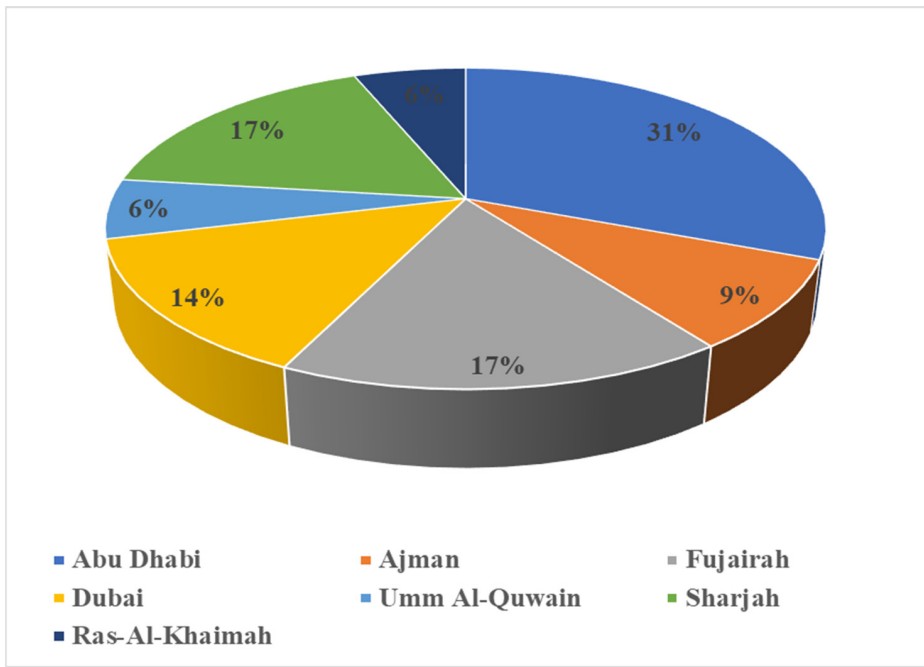

**Figure 3.** Percentage of wastewater treatment sites in the United Arab Emirates (Data: Federal Competitiveness and Statistics Authority, 2017).

## 7. Treated Wastewater Irrigation in UAE

The total agricultural land used for irrigation with treated (partly treated, treated, or diluted) wastewater was about 20 million ha [52–94]. Mohammed Qadir et al. [92] reported that a significant increase in crop yields was observed from treated wastewater irrigation rather than groundwater or freshwater irrigation. However, in recent years, the agricultural use of wastewater to vegetation was the last option because this can increase humans' environmental and health risks [15,93]. The World Health Organization (WHO) set standards and revived specifications with the FAO for wastewater reuse in agriculture [95].

Furthermore, these organizations suggested that the standards be followed for water types, especially in water-shortage countries. The purpose of a high degree of control for wastewater reuse is to make recommendations that the water contains the right amount of nutrients and is deemed acceptable. Therefore, they established a policy framework, enforcement mechanisms, strong regulations, and an institutional setup. Many GCC countries updated their water resources and established specifications for the treated wastewater plant. It was recommended that wastewater reuse be adapted to the following public sectors; parks, gardens, and landscaping. However, only about seven percent of wastewater finds its use in agriculture [96] and 30% in the public sector. The disposal of wastewater into the sea is causing environmental pollution [97].

Because of the scarcity of cultivatable land in the metropolitan cities, attention was diverted to studying the soilless cultivation of crops. Therefore, plant physiologists need to know that plants can digest heavy metals and acclimatize the metal. The methods mentioned above indicate that the root system should not be left in a dry condition. Ghaly et al. [98] showed that forage wheat grown under soilless treated wastewater irrigation system exceeded the soil standard test water system of alfalfa crops by ninety-eight-fold under irrigation with wastewater.

## 8. Conclusions and Future Perspectives

The UAE long tried to achieve food self-sufficiency, but demand for food imports appears to be increasing at a rate of about 0.80 percent per year, faster than population growth, which will inevitably stymie the country's economic development in the near future. Using treated wastewater effectively in agriculture and other sectors could help

Arab countries minimize water scarcity and pay for food security. The study analyzed a number of methods for treatment that can be applied to wastewater recycling. The Arab nations provided two possibilities for reusing wastewater, which includes a decision that would set the standard of the treated wastewater and the technology utilized to treat the wastewater. This is believed to increase the flexibility of the reuse of the treated wastewater for various purposes. The other way, an existing approach, is that the quality of the treated wastewater determines the effluent's use. Much more consideration needs to be put into adopting treatment methods in agricultural fields. However, the use of treated wastewater in agriculture should ensure the effluent's quality for which effective treatment technologies have to be practiced, and reuse management protocol should be standardized.

**Author Contributions:** Conceptualization: M.A.M.A.; Formal analysis: F.H.A.H. and K.K.; Investigation: F.H.A.H. and M.R.; Methodology: F.H.A.H. and K.K.; Project administration: M.A.M.A. and A.J.; Supervision: M.A.M.A.; Validation: Y.L. and M.R.; Writing—original draft: F.H.A.H., A.J. and K.K.; Writing—review & editing: Y.L. and M.A.M.A. All authors have read and agreed to the published version of the manuscript.

**Funding:** This research received no external funding.

**Data Availability Statement:** Data sharing is not applicable.

**Conflicts of Interest:** The authors declare no conflict of interest.

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
