# Peer review of "Wastewater Irrigation: A Promising Way for Future Sustainable Agriculture and Food Security in the United Arab Emirates"

_water, doi:10.3390/w15122284_

Round 1

Reviewer 1 Report

The manuscript deals with a crucial topic and is a time-worthy work indeed. However, some points need to re-address to improve the manuscript.

Author Response

Please find attached the file for the reviewer's response for your kind inspection.

Reviewer 2 Report

General Comments:

The manuscript titled “Wastewater irrigation: A promising way for future sustainable agriculture and food security in the United Arab Emirates” has great potential and some enlightening significance about the impact of wastewater irrigation on agriculture. Giving readers and researchers a new perspective to be concerned about the effects of wastewater irrigation, this is a place worth learning from. But there are few typing and grammatical mistakes in the manuscript. I suggest authors to review your document completely by a native English speaker to resolve these shortcomings. Moreover, there are many important numerical figures, those are not supported with references. The specific comments are listed below:

Specific Comments:

·       Line 20-21: the sentence “The rapidly increasing world population rate will undoubtedly demand an increased food production rate, which directly impacts agricultural water usage.” should be reworded as “The rapidly increasing world population will undoubtedly increase the food demand, which directly requires more water for irrigation purposes.” Or other suitable words.

·       Line 23-24: “……on summarising all the leading researches in the field” the word researches should be replaced with “studies”.

·       Keywords are important to make your article more discoverable in online searches. Keywords should not be overlapped with the title. Please see the journal'js requirement and replace the “wastewater irrigation” with other suitable word, if this word isn’t really important.

·       Line 40: “……and development of rainfall infrastructure development projects,” the repetition of “development” word could confuse the readers.

·       Figure 1 and 4: authors are requested to please make sure there is no copy right issue for these figures.

·       Line 74-75: “It is estimated that by 2050 the water requirement per capita in the MENA region will fall by half.” Authors are requested to please support this statement with a valid reference.

·       Line 82: “Among the counties…….”, the word “counties” must be replaced with “countries”.

·       Line 90: “Environment Agency - Abu Dhabi”, is it possible to provide the full reference, any published report, book, or blog.

·       Line 92: The word “absorbs” looks inappropriate here, please replace this with a relevant one. Moreover, it would be appreciated, if the statistics data i.e. “550 liters of water per person per day” supported with reference.

·       While assigning the acronyms to the words, the initial letters of those words must be mentioned while using the words first time i.e. in Line 100, authors used “TWW” with mentioning its full form. Similarly, in Line 112 “MCM”, authors can assign this acronym in Line 100 as “groundwater contributed 2218 million m3 (MCM)”. Please rewrite this and use the same format throughout the whole manuscript.

·       Line 108-109: Please support “Agriculture is a sector that consumes nearly 70% offreshwater” with reference.

·       Line 114: The cited references [21,22] seem to be invalid, authors are requested to re-check these.

·       Section: 2. Wastewater and its treatment, authors spend a lot in explaining the wastewater treatment processes i.e. primary, secondary, and tertiary. According to title and study objectives these are not really important, if these are, authors are requested to make symmetry of this section with study objective.

·       Section: 3. Treated wastewater in agriculture, this section is very important and could be the backbone of this study. However, unfortunately authors are still focusing and discussing the importance (line 277-278), origin (line 274-275, 294-295), and health risks (line 260-262) of wastewater. I believe this is the place where authors need to discuss the results from literature, that how wastewater irrigation impact the crop productivity, save the fresh water, and impact of wastewater irrigation on soil health supported with relevant refences, although authors did a good job but needs more modification.

·       Authors are requested to improve this section according to the study objective and try to make a good symmetry/alignment among the paragraphs.

·       Line 318: In sentence “The WHO's standards on soil irrigation with the treated wastewater” the “soil irrigation” seems inappropriate. Authors again use this term in Line 400. This term can confuse the readers, authors are requested to briefly highlight more this term for better understanding.

·       Authors are repetitively explaining the importance of wastewater, and global food scenarios in every section i.e. Line: 398-400, 405-411, although these are the introductory lines, and authors already explained this in beginning section.

·       Sub-section “1.2 Water scarcity in the UAE” and section “4. Water scarcity for agriculture in the UAE” are overlapped.

·       The conclusion should be itemized. Please arrange conclusion according to the objectives of the study. Could the authors use the conclusions to look at the broader implications for regions outside this specific area? There must be a few things in here that will inform studies in other regions that will give the paper more impact.

There are few typing and grammatical mistakes in the manuscript. I suggest authors to review your document completely by a native English speaker to resolve these shortcomings. Detailed comments could be found in "specific comments" section.

Author Response

(The authors gave the same response as above.)

Round 2

Reviewer 1 Report

The revised manuscript does meet the normal standards of writing a scientific article for a research paper. 

I recommend the manuscript for publication in the present form.

Keep on your good work.